# Safety and Efficacy of Second Ahmed Valve Implant in Refractory Glaucoma

**DOI:** 10.3390/jcm9072039

**Published:** 2020-06-29

**Authors:** Chiara Posarelli, Mario Damiano Toro, Robert Rejdak, Tomasz Żarnowski, Dorota Pożarowska, Antonio Longo, Mario Miccoli, Marco Nardi, Michele Figus

**Affiliations:** 1Department of Surgical, Medical and Molecular Pathology and of the Critical Area, University of Pisa, 56126 Pisa, Italy; marco.nardi@med.unipi.it (M.N.); michele.figus@unipi.it (M.F.); 2Department of General Ophthalmology and Pediatric Ophthalmology Service, Medical University of Lublin, 20079 Lublin, Poland; toro.mario@email.it (M.D.T.); robertrejdak@yahoo.com (R.R.); 3Faculty of Medical Sciences, Collegium Medicum Cardinal Stefan Wyszyński University, 01815 Warsaw, Poland; 4Department of Diagnostics and Microsurgery of Glaucoma, Medical University, 20079 Lublin, Poland; zarnowskit@poczta.onet.pl (T.Ż.); dpozarowska@wp.pl (D.P.); 5Eye Clinic, University of Catania, 20079 Catania, Italy; antlongo@unict.it; 6Department of Clinical and Experimental Medicine, University of Pisa, 95123 Pisa, Italy; mario.miccoli@med.unipi.it

**Keywords:** refractory glaucoma, Ahmed Glaucoma Valve implantation, intraocular pressure

## Abstract

Background: Refractory glaucoma still represents a challenge for ophthalmologists to manage intraocular pressure. The present study aimed to evaluate long term efficacy and safety of a second Ahmed valve implantation after the failure of a first implant in patients with refractory glaucoma and elevated intraocular pressure (IOP). Methods: Retrospective, multicenter non-comparative case series. Twenty-eight patients were retrospectively recruited between January 2011 and December 2017. Demographic data, glaucoma type, visual acuity, intraocular pressure, medical therapy, and complications were registered. Three criteria of success were established: Type 1 surgical success: IOP ≤ 15 mmHg and a reduction of IOP ≥ 40% from baseline; Type 2 surgical success: IOP ≤ 18 mmHg and a reduction of IOP ≥ 30% from baseline; and Type 3 surgical success: IOP ≤ 21 mmHg and a reduction of IOP ≥ 20% from baseline. Surgical failure has been established as IOP less than 5 mmHg or over 21 mmHg and less than a 20% reduction of IOP from baseline despite medications in two consecutive visits, light perception loss referable to glaucoma, and the necessity for further glaucoma surgery. Failure was observed in six (21%) patients. (3) Results: Mean IOP and mean glaucoma medication number significantly reduced from baseline after the second implantation, and the surgical success rate at 72 months ranged from 10% to 78% based on the different criteria of success. Failure was observed in six (21%) patients. Conclusions: This study confirmed the safety and efficacy of a second Ahmed valve implantation in patients with refractory glaucoma and elevated IOP at baseline.

## 1. Introduction

Glaucoma represents the second cause of vision loss worldwide, and the management of glaucoma patients may benefit from surgery to preserve visual function [1]. Refractory glaucoma cases are characterized by advanced optic nerve damage, severe impairment of visual field, and a non-controlled intraocular pressure (IOP) despite maximal medical therapy and previous glaucoma surgeries. In these cases, the management of IOP may be difficult, but preservation of visual function should guide our treatment choices. Multiple surgeries may be required due to the high rate of surgical failure and inadequate response to traditional surgical and medical treatments [2,3]. Almost all types of glaucoma may evolve to refractory glaucoma, and the state of the conjunctiva together with a patient’s systemic health state may precipitate this condition. For these reasons, glaucoma filtration devices such as the Ahmed Glaucoma Valve (New World Medical, Rancho Cucamonga, CA, USA), Baerveldt (Advanced Medical Optics, Inc., Santa Ana, CA, USA), or Molteno (IOP, Inc., Costa Mesa, CA, USA, and Molteno Ophthalmic Limited, Dunedin, New Zealand) valves have been demonstrated to be safe and effective in lowering IOP and are therefore widely used in complex cases such as secondary and refractory glaucoma [4,5,6,7,8,9]. 

The Ahmed Glaucoma Valve (AGV) presents the advantage of flow-restricting technology. Inside the plate of the device, there is a Venturi valve, which lets the flow through only for specific IOP values. This mechanism avoids hyperfiltration and early hypotony. Compared to non-valved glaucoma drainage implants, AGV displayed fewer complications related to the surgical technique and hyper-filtration in the first postoperative period. Typically, non-valved devices present no aqueous humor resistance and the flow should be controlled by placing absorbable Vicryl^®^ (Ethicon Inc., Bridgewater, NJ, USA) sutures around the tube, or directly obstructing the lumen of the tube with a nylon or prolene suture, or performing a two-step surgery with the tube placed under the conjunctiva in the initial phase and after one month in the anterior chamber [3]. Furthermore, the valve could be correctly positioned and fixed to the sclera without interference with the rectus muscles [3]. 

The success rate of the AGV implant in refractory and complicated cases (secondary glaucoma: neovascular, uveitic) varies from 60% to 82% at two years follow-up. It reduces to 49% at five years after surgery, with a 10% per year failure rate [3,8]. Dubey and co-authors reported a success rate in Northern Indian eyes with refractory glaucoma of 85.45% at one year and 79.63% at three years after AGV implantation [9]. In the case of AGV implant failure, the possible subsequent options are the revision of the first AGV (tube repositioning and capsule revision), a second AGV implant, or cyclodestructive procedures. Since cyclodestructive procedures may lead to severe complications such as flattened anterior chamber, hypotony, and ocular phthisis [10], the implantation of a second AGV could represent a valid alternative. 

The aim of this study was therefore to evaluate the efficacy and safety profile of second AGV implants in patients affected by refractory glaucoma with elevated IOP.

## 2. Methods

### 2.1. Study Design and Participants Selection

Retrospective, multicenter non-comparative case series. Patients who underwent a second AGV implant were retrospectively recruited at the University of Pisa, Italy, and at the Department of Diagnostics and Microsurgery of Glaucoma at the Medical University of Lublin (Poland). Patients under 18 years of age at the time of the second implant were excluded from the study population as long as patients had less than three months of follow-up after surgery.

### 2.2. Ethics Statement

The study design was approved by the Ethics Committee of the Medical University of Lublin, Poland, and by the Ethical Committee Area Vasta Nord-Ovest (CEAVNO) in Pisa (KE-0254/117/2020). The study design was conducted according to the principles of the Declaration of Helsinki (1975) and its revised version of 2013. The written informed consent was not necessary based on the indications of the above-mentioned institutional review boards because there was no conflict with the clinical practice. Data were anonymized for collection and analysis, avoiding a privacy data breach. 

### 2.3. Data Collection and Outcome Measures

Medical databases of the two centers were searched between January 2011 and December 2017 (six years). For study purposes, the demographic data, glaucoma type, best corrected visual acuity (BCVA), IOP (measured with calibrated Goldmann applanation Tonometer), medical therapy, and complications were registered and statistically elaborated at each time point. Visual field data were not collected because they were not available for the majority of patients or because they were not carried out for low vision.

Surgical success has been defined according to the World Glaucoma Association’s guidelines [11] as follows:

Type 1 surgical success: IOP ≤ 15 mmHg and a reduction of IOP ≥ 40% from baseline;

Type 2 surgical success: IOP ≤ 18 mmHg and a reduction of IOP ≥ 30% from baseline; and

Type 3 surgical success: IOP ≤ 21 mmHg and a reduction of IOP ≥ 20% from baseline.

Surgical failure has been defined as the presence of at least one of the following: IOP less than 5 mmHg or over 21 mmHg and less than a 20% reduction of IOP from baseline despite medications in two consecutive visits, light perception loss referable to glaucoma, and the necessity for further glaucoma surgery (needling and bleb revision were not considered as failure).

Safety parameters registered were the best corrected visual acuity measured with the logarithm of the minimum angle of resolution (LogMAR) charts, early and postoperative complications, need for implant removal, and need for other surgical treatments.

### 2.4. Surgical Technique

The second Ahmed Glaucoma Valve implantation was performed using a standard technique. A fornix-based conjunctival flap was sculpted in the superonasal or inferotemporal quadrant, based on the availability of healthy conjunctiva and the surgeon’s preference. Both surgeons avoided the applications of subconjunctival antimetabolites. Subsequently, the Ahmed valve was primed, and its plate was positioned 10–11 mm away from the limbus and sutured to the sclera with nylon 8–0 suture. Then, the AGV tube was cut beveled up and placed into the anterior chamber. The anterior portion of the tube was fixed to the sclera by applying nylon 10–0 sutures and covered with donor sclera or a patch of bovine pericardium membrane, Tutopatch^®^ (Tutogen Medical GmbH, Neunkirchen am Brand, Deutschland). Finally, the conjunctiva was secured with Vicryl^®^ 7–0 single stitches. To reduce variability, one expert surgeon (T.Ż. and M.N.) for each center performed all surgical procedures. Postoperatively, all patients were prescribed a fixed combination (netilmicin 0.1%–dexamethasone 0.3%), five times a day for four weeks, which was tapered over the next 6–8 weeks. 

Patients were examined on the postoperative day (day 1), then the day after (day 2), and every two weeks for three months (week 2, week 4, week 6, week 8, week ten, and week 12), then after four months (week 16), six months, twelve months, and every six months till the end of follow-up, and every time it was needed, according to the standard used protocols.

### 2.5. Statistical Analysis

The Shapiro–Wilk test was used to verify the normality of distributions. Analysis of variance (ANOVA) for repeated measures with the Dunnet procedure for multiple comparisons with the basal group was performed to analyze the Gaussian variable. Friedman’s test with Shaffer’s extension was carried out to study the factor not normally distributed. 

Kaplan–Meyer functions were used to describe the evolution of the successes during the follow-up. The mean success time, the standard deviation, and the 95% CI of the mean were reported to show the trend.

The two one-sided test (TOST) procedure for repeated measures was performed to evaluate the equivalence of visual acuity (logMAR) between the time points. In this case, the intraclass correlation coefficient (ICC) was also calculated to estimate the inter-rater agreement measures at different times point. Values of ICC between 0.8 and 1 were considered excellent levels of agreement. 

The significant level of the tests was set to 0.05, and the statistical analysis was carried out with R 3.5.0. and XLSTAT 2018.

## 3. Results

Twenty-nine patients (17 men and 12 women) who underwent two consecutive AGV implants during the study period were identified. One male patient was lost before three months of follow-up; the patient died of heart failure, so twenty-eight patients were finally enrolled. The mean age of our population was 59 years (range 26–79 years), any patient was excluded for age criteria. The main clinical characteristics of the patient population after first implant failure are summarized in Table 1; primary open-angle glaucoma (18%), pseudo-exfoliative glaucoma (18%), and congenital cataract glaucoma (14%) were the most represented glaucoma type in our series. Other glaucoma types in our series were primary angle-closure glaucoma (7%) and secondary glaucoma (11%) (uveitic, neovascular, post-traumatic, congenital cataract, and silicon-oil related glaucoma).

All patients had at least one surgical procedure before the first Ahmed Glaucoma Valve implantation and twenty-two patients (79%) before the second implant already had two surgical glaucoma interventions. No patient had AGV as the primary glaucoma surgery.

The mean time between the first and the second AGV implant was 1.8 ± 2.1 years (range three months–8 years).

After the second AGV implantation, the mean intraocular pressure and mean glaucoma medication number significantly reduced from the baseline. Preoperatively, the mean IOP was 35.2 ± 7.9 mmHg and lowered significantly to 15.5 ± 3.4 (*p* < 0.01) after 12 months, to 15.44 ± 42 (*p* < 0.01) after 24 months, to 13.2 ± 3.5 (*p* < 0.01) after 36 months, to 15 ± 2.6 (*p* < 0.01) after 48 months, to 14.4 ± 3.3 (*p* < 0.01) after 60 months, and to 13.4 ± 3.9 (*p* < 0.01) after 72 months, respectively (Figure 1). 

Mean number of medications was 2.7 ± 0.8 (range 1–4) and was reduced significantly postoperatively to 1.2 ± 1.1 (*p* < 0.01) at 12 months; to 1.5 ± 1.1 (*p* < 0.01) at 24 months, to 1.6 ± 1.1 (*p* < 0.05) at 36 months, to 1.3 ± 1.3 (*p* = 0.05) at 48 months, to 1.0 ± 1.3 (*p* = 0.06) at 60 months, and to 1.2 ± 1.3 after 72 months (*p* = 0.08) (Figure 2).

Surgical success rates at 12, 36, 60, and 72 months after a second AGV implant, according to Type 1 criteria of success (IOP ≤ 15 mmHg and a reduction of IOP ≥40% from baseline) were 89% ± 0.06, 65% ± 0.09, 58% ± 0.1, and 10% ± 0.1; for Type 2 criteria of success (IOP ≤ 18 mmHg and a reduction of IOP ≥30% from baseline) they were 92% ± 0.05, 80% ± 0.08, 72% ± 0.1, and 24% ± 0.1; and for the Type 3 criteria of success (IOP ≤ 21 mmHg and a reduction of IOP ≥20% from baseline), they were 95% ± 0.05, 86% ± 0.07, 78% ± 0.1, and 77% ± 0.1, respectively (Figure 3).

The mean time of survival was 39. 4 months (range 34.4–44.4) for Type 1 criteria of success; 42.6 months (range 38.1–47.2) for Type 2 criteria of success, and 45 months (range 41.3–48.7) for Type 3 criteria of success. Tonometric failure was observed in six patients (21%). Particularly, two patients developed transient hypotony, respectively 24 and 30 months after the second AGV. Mean visual acuity at baseline was 1.1 ± 0.8 LogMAR, and remained stable, respectively, after 12 (1.2 ± 0.9 LogMAR), 36 months (1.2 ± 1 LogMAR), and 72 months (1.2 ± 0.9 LogMAR) postoperatively (TOST procedure for repeated measures: *p* < 0.001; interclass correlation coefficient 0.9; *p* < 0.001); one eye lost light perception after the second AGV implant. In terms of safety, early and late complications were registered and shown in Table 2. None of the patients had serious intra operative complications that led to loss of vision. In the early postoperative period, seven patients (25%) had transient IOP spikes; less frequently observed were transient hypotony, hyphema, serous choroidal detachment, and flattened anterior chamber. Lately, patient complications have been corneal edema, implant exposure, and prolonged hypotony. One patient had endophthalmitis that was solved with medication. Motility disorders were not reported in our series.

In our series, further surgical interventions performed after the second AGV implant were tube shortening (four patients), capsule revision (three patients) without the use of antimetabolites, and implant removal (three patients). In addition, other surgeries executed were: Descemet stripping automated endothelial keratoplasty (DSAEK) (three patients), anterior vitrectomy (two patients), anti-VEGF intravitreal injections (two patients), and cyclophotocoagulation (one patient).

## 4. Discussion

This study aimed to assess the safety and efficacy of a second Ahmed Glaucoma Valve implant in refractory glaucoma patients with elevated baseline IOP already treated with a first AGV implant. These patients represent a challenge for ophthalmologists, and traditional medications and surgical procedures have failed to achieve an adequate IOP value. 

The mean IOP of our population at baseline was high: 35.2 ± 7.9 mmHg. Despite maximum medical therapy and previous surgery, 25% of patients had an IOP between 25 and 30 mmHg; 39% of patients had an IOP between 30 and 40 mmHg, and 21% of patients had an IOP between 40 and 50 mmHg, two patients had an IOP < 21 mmHg, and one > 50 mmHg. All patients had at least one surgical procedure before the first Ahmed Glaucoma Valve implant and twenty-two patients (79%), before the second implant had already had two surgical glaucoma interventions. No patient in the present series had AGV as primary glaucoma surgery. Frequently, cyclodestructive procedures represent an alternative treatment option in these cases, but with an elevated risk of severe complications such as prolonged hypotony, malignant glaucoma, and ocular phthisis [12,13,14,15,16,17,18]. A second AGV implantation could instead represent an efficient surgical option. Both valved or non-valved glaucoma drainage implants have been used in patients with refractory glaucoma with a variable degree of success. 

Our population had a very high IOP at baseline, and there was a real risk of hyperfiltration and hypotony. For these reasons, in both centers, devices with flow-restricting technology have been preferred. In terms of efficacy, our results are in line with those already published in the literature [19,20,21,22,23,24,25,26,27,28]. The success rate reported in a recent metanalysis of Yoon et al. [29] ranged between 37% to 83% at three years with a significant reduction in glaucoma medications. Considering the baseline IOP of our patients, the target IOP after a second AGV implant should be <15 mmHg with a lowering of IOP ≥ 40% from the baseline, and according to the Type 1 criteria of success, our rates were 89% after 12 months, 65% after 36 months, 58% after 60 months, and 10% after 72 months, and confirmed previous pieces of evidence [29]. Success rates in our series may have been influenced by the sample size and by baseline tonometric values that were higher than those reported in the previous one [29]; IOP at baseline was > 21 mmHg in 26 out of 28 patients with a mean value of 35.2 ± 7.9 mmHg. Consequently, our findings seem to confirm the observation of Smith et al. that for patients with higher IOP at baseline, a second AGV implant obtained better IOP lowering effect [23]. Indeed, we observed an inverse relationship between the baseline and final IOP values: patients with higher IOP at baseline were those with the lowest values at the last follow-up visit. Reducing visual field progression and maintaining visual function is fundamental to the achievement of target intraocular pressure (Figure 4).

The study from Fatehi et al. presented data after a second AGV with longer follow-up times. [28]. The authors reported the results of a second AGV implant in 110 eyes of 104 patients followed for five years after surgery. Success was defined using three stringent criteria, based on target IOPs: A-IOP < 21 mmHg and 20% IOP reduction from baseline; B-IOP < 18 mmHg and 25% IOP reduction from baseline, or C-IOP < 15 mmHg and 30% IOP reduction from baseline. Failure was considered in the case of implant removal, vision loss, or missing success criteria in two consecutive visits. According to the criteria above-mentioned, the success rate at 1, 3 and 5 years after surgery was 70%, 62%, and 56.9%, respectively for criteria A; 64.8%, 55.4%, and 51.1% for criteria B; and 50.6%, 36.2%, and 30.3% for criteria C. In our series, we experienced a higher rate of success, probably due to the higher IOP at the baseline of our population (35.2 ± 7.9 mmHg vs. 25.72 ± 9.27 mmHg of Fatehi’s study), as we have previously discussed.

Shaefer et al. in 2015 published another report with even a longer follow-up time after a second glaucoma drainage device implant [30]. These authors compared cyclophotocoagulation with a second drainage implant. However, considering only data regarding the success of a second valve, they observed that after 11 years, nine out of 15 patients failed, but only three before four years. The authors found that 60% of patients failed for an inadequate IOP lowering effect, and the majority failed in the fifth year (56%), so they hypothesized that the aqueous and tissue risk factors for failure were the same of the first glaucoma implant, and were more evident with longer follow-up time. The second drainage glaucoma implants in this study were Baerveldt 350 mm^2^ and Molteno valve and not AGV; consequently, these results are less comparable to ours than those of Fatehi [28], who used only AGV implants.

Moreover, considering our results, the mean time of failure ranged between 39 to 45 months after surgery; these differences may be related to the elevated IOP values at baseline and also to the stringent criteria of success that we established. Finally, in the previous study [30], the reduction of glaucoma medications did not reach statistical significance at the final follow-up visit, and these results are concordant with those of Godfrey [21]. Still, there have been other reports [23,24,25,26,27] of a significant decrease in glaucoma medications. Based on our findings, we observed a statistically significant reduction of glaucoma medications until month 60, but not at the final follow-up visit. The small sample may also explain our results with a follow-up of six years after the second AGV implant.

Additionally, all our patients, after the failure of the first AGV implant, received a second AGV implant and no other devices. Recently, Ong et al. published a report regarding the use of a second AGV implantation in Asian eyes with refractory glaucoma. Authors reported in this Asian population, a failure rate of 9.5%, 20.0%, 32.5%, and 46% at six months, one, two, and three years of follow-up, respectively; the reduction of IOP and medications was statistically significant at the final follow up visit. These results are concordant with our findings in Caucasian descent. Still, again the baseline IOP values of our population were higher (35.2 ± 7.9 mmHg vs. 23.4 ± 7.9) and our criteria of success were more stringent when compared to the one used by the authors. The first glaucoma drainage implant represents another difference because in our series, all patients underwent AGV, instead, in the paper published by Ong and co-authors, patients also underwent a Baerveldt implant in one case and in one patient it was not reported; moreover, looking at the glaucoma types, no cases of primary angle closure glaucoma were included despite the Asian race and primary open angle glaucoma cases being the most frequent [31]. Finally, regarding visual acuity during follow-up, this group validated our findings, but no significant difference in terms of BCVA was reported [31].

Our results confirmed the safety of a second AGV implant. Corneal decompensation is the most common complication described in the literature, followed by IOP spike, shunt exposure, suprachoroidal hemorrhage, and vitreous hemorrhage [32]. In our series, we had eight patients (29%) who developed corneal edema, and three patients needed endothelial keratoplasty. Two of them had already had a lamellar keratoplasty before the second AGV implantation. Tube-corneal touch was the leading cause of decompensation, and for this reason, tube shortening was performed in four patients, along with follow-up. Intraocular pressure spikes (25%) coincided with the hypertensive phase and could be related to the encapsulation of the bleb [26,31]. Three patients (11%) in our series developed a fibrotic capsule around the plate of the valve and did the revision. Our patients did not report motility disorders. A possible explanation could concern the surgical technique and the placement of the valve without interference with the rectus muscles.

Implant removal was performed in three cases. Two of them were patients with congenital cataract with very high IOP at baseline. The same patients also underwent anterior vitrectomy before implant removal. The last case developed corneal decompensation 18 months after AGV implant and after tube shortening, the implant was removed. Moreover, implant removal was justified only in cases of chronic implant exposure with the inability to cover it, because this maneuver is technically challenging, even for expert surgeons, and could especially present elevated risks of suprachoroidal hemorrhage and infection. One patient had endophthalmitis that was solved with medications, and one had cyclophotocoagulation. In this case, glaucoma was related to silicone oil in the anterior chamber, and after two revisions, IOP was still high, and the visual acuity worsened. Visual field data were not available for the majority of our patients or were not performed due to low vision. Nevertheless, visual acuity equivalence from the baseline to final follow up visit proved the maintenance of the patient’s visual function. 

Based on our findings, and supported by previously published literature, in patients with residual visual acuity, both a second glaucoma drainage device implantation or cyclodestructive procedure may be useful in reducing IOP with a sufficient safety profile [15,33,34]. The results of the running randomized clinical trial of the American Glaucoma Society (AGS): Second Aqueous Shunt Implant versus Transscleral Cyclophotocoagulation Treatment Study (ASSISTS, NCT02691455), might indicate the better treatment option after a first failed tube implant. 

The strength of our study is represented by the stringent IOP criteria of success based on baseline intraocular pressure values, as reported by the World Glaucoma Association’s guidelines [10]. Target postoperatively intraocular pressure depends on baseline IOP values; the higher the values at baseline, the lower they should be after surgery. Visual function maintenance relies on the achievement of target IOP and consequently on the reduction of visual field progression. The main limitations of the present study are the retrospective and non-comparative design, moreover, the relatively small sample size, and the non-uniformity of the population may represent other biases. To evaluate the sample sizes, post-hoc power analyses were performed and may justify the loss of statistical significance at final follow-up visits for outcome measures. Nevertheless, based on published evidence and on our results, a second AGV implant could be offered in refractory glaucoma patients with elevated IOP with a good chance of intraocular pressure lowering. 

## 5. Conclusions

The results of our retrospective study confirmed the safety and efficacy of a second Ahmed Glaucoma Valve implantation in patients with refractory glaucoma and elevated IOP at baseline. A reduction of IOP associated with fewer glaucoma medications and preserved visual acuity was observed across the entire study duration. Further prospective and comparative studies will be necessary to confirm our findings.

## Figures and Tables

**Figure 1 jcm-09-02039-f001:**
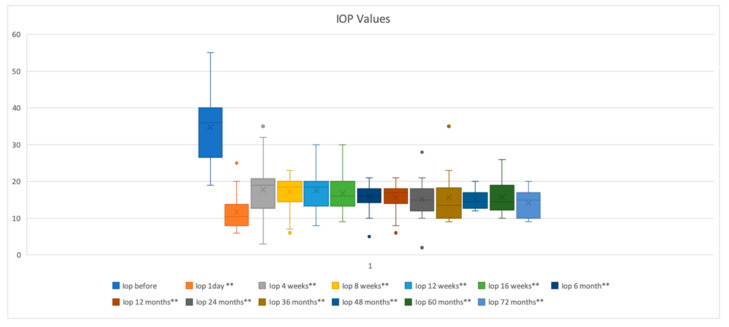
Intraocular pressure (IOP) during follow up. Representation of mean IOP using boxplots for 72 months of follow-up, IOP was significantly reduced at each time point (** *p* < 0.01). Every time point is represented as the “minimum” and the “maximum” value, the mean value, the median value, and the outliers (color dots) graphically.

**Figure 2 jcm-09-02039-f002:**
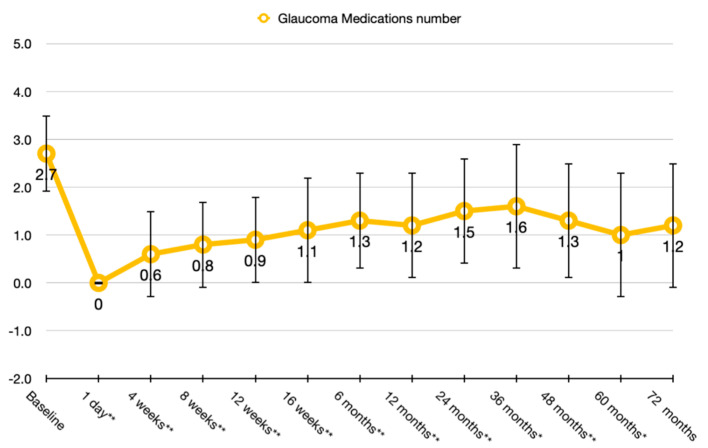
Glaucoma medication trend after the second Ahmed Glaucoma Valve (AGV) implant. Representation of the number of glaucoma medications. The reduction was statistically significant (** *p* < 0.001; * *p* < 0.05) at each time point until month 60.

**Figure 3 jcm-09-02039-f003:**
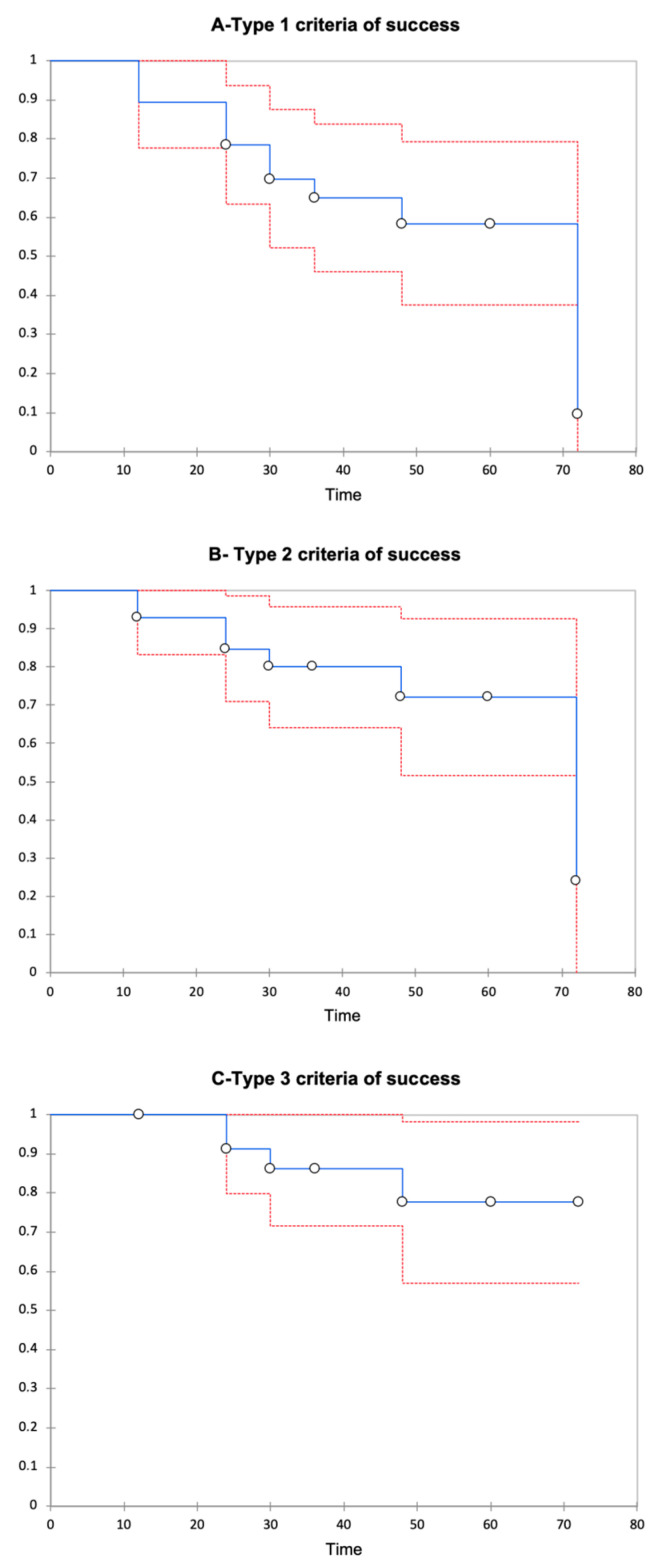
Kaplan–Meier survival curve in all populations. Kaplan–Meier survival curve, showing the success rate of a second Ahmed Glaucoma Valve implantation in subjects with previous failure of the AGV in the same eye. The success rate of the second AGV was calculated based on the criteria of success Types 1 (**A**), 2 (**B**), and 3 (**C**); time is expressed in months.

**Figure 4 jcm-09-02039-f004:**
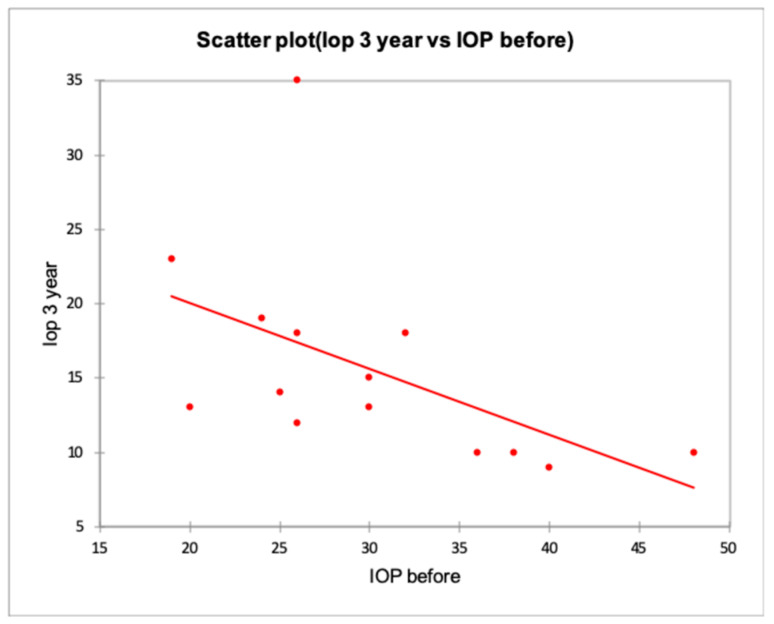
The inverse relationship between the baseline and final IOP values at 36 months. The image showed an inverse relationship between IOP values before and after surgery. Higher IOP at baseline resulted in lower IOP after surgery.

**Table 1 jcm-09-02039-t001:** Demographic and clinical characteristics of the patient population.

Mean Age	59 Years (Range 26–79 Years)
Glaucoma Type	Pseudo-exfoliative glaucoma (5 pts) Congenital Cataract glaucoma (4 pts) Neovascular glaucoma (3 pts) Uveitic glaucoma (3 pts) Post-traumatic glaucoma (3 pts) Silicone-oil related glaucoma (3 pts) Primary Open Angle Glaucoma (5 pts) Primary Angle Closure Glaucoma (2 pts)
Mean IOP ± SD	35.2 ± 7.9 mmHg (range 19–55 mmHg)
Mean number of anti-glaucoma medications ± SD	2.7 ± 0.8 (range 1–4)
Mean number of previous glaucoma surgeries (including first AGV implant) ± SD	3.68 ± 2.6 (range 2–11)
Mean time between first and second AGV implant ± SD	1.8 ± 2.1 years (3 months–8 years)

IOP: intraocular pressure, AGV: Ahmed glaucoma valve; pts: patients, SD: standard deviation.

**Table 2 jcm-09-02039-t002:** Postoperative complications after second Ahmed Glaucoma Valve implantation.

Postoperative Complications		*n* (%)
Early postoperative complications	IOP Spikes	7 (25%)
	Transient hypotony	4 (14%)
	Hyphema	2 (7%)
Late postoperative complication	Corneal edema	8 (29%)
	AGV exposition	1 (4%)
	Endophthalmitis	1 (4%)
	Prolonged Hypotony	1 (4%)

*n*: number of patients; % percentage.

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
