# Peer review of "Safety and Efficacy of Second Ahmed Valve Implant in Refractory Glaucoma"

_jcm, 2020, doi:10.3390/jcm9072039_

Round 1

Reviewer 1 Report

Posarelli et al have conducted a retrospective study on 28 participants with refractory glaucoma and having elevated baseline IOP already treated with a first AGV 161 implant.

Importantly, there are several more concerns with the paper and analyses that need to be addressed: 

Plagiarism check showed a level of 28%, which is quite high. This level of overlap is unacceptable, however, if authors would like their article to be considered for publication, they need to make a thorough revision to keep the Similarity Index ≤ 20% and re-submit it, addressing these concerns. Specifically, the methods section needs rephrasing. 

Abstract section:

Lines 17-19: The mentioned lines are the aim of the study, however, it falls under the background. The authors are encouraged to either replace the term background with the term aim or very briefly add a line on the background of the study.  

Methods section: 

Methods section needs to be restructured: 

Line 53: the heading experimental section is unfamiliar and needs to be replaced with Methods.

Lines 54-58: These lines should fall under the subsection Ethics Statement.

Line 58: Institutional review board number is missing and needs to be added. 

Lines 59-93: these lines need are not organized and need to  fall under the subsections such as patient recruitment, surgical technique, data collection... Authors are encouraged to reformat this part of the methods. 

Results section:

The comma in decimals values should be replaced with ‘.’ all over the text. 

Line 120: IOP -> needs to be replaced with its acronym. Figure caption is missing; authors are encouraged to briefly describe the figure (box plots), the type of test used, etc.. 

Line 128: Idem as Figure 1.

Line 185: Idem as Figure 1.

Discussion Section: 

Searching the literature identified several similar articles: 

Safety and Efficacy of Ahmed Glaucoma Valve Implantation in Refractory Glaucomas in Northern Indian Eyes. Suneeta Dubey et al, 2015. 

Surgical Outcomes of a Second Ahmed Glaucoma Valve Implantation in Asian Eyes with Refractory Glaucoma. Sze Chuan Ong et al, 2020 

Authors need to compare their results to these articles and mention their novel and the added value of their current manuscript.  

Line 164-167: this phrase is misplaced; authors are encouraged to move it to the results section.

Lines 235-237: the ID of the clinical trial needs to be added: NCT02691455. 

Line 241: As mentioned in the limitation, the authors have conducted their study on a limited sample size... Did they try to carry out any sampling/experimental design before conducting the study? Determining the optimal sample size for their study could provide readers about the adequate number of participants needed to detect robust significant results! 

Author Response

Posarelli et al have conducted a retrospective study on 28 participants with refractory glaucoma and having elevated baseline IOP already treated with a first AGV 161 implant. Importantly, there are several more concerns with the paper and analyses that need to be addressed:

Plagiarism check showed a level of 28%, which is quite high. This level of overlap is unacceptable, however, if authors would like their article to be considered for publication, they need to make a thorough revision to keep the Similarity Index ≤ 20% and re-submit it, addressing these concerns. Specifically, the methods section needs rephrasing.

We thank the reviewer for the comment, the plagiarism document that you received included references, we directly contacted the Assistant Editor Alina Leng for a new check and after removing references the level decreased to 22%. We also modified the sentences with high repetition rate along the manuscript as suggested. The all method section has been rephrased

Abstract section:

Lines 17-19: The mentioned lines are the aim of the study; however, it falls under the background. The authors are encouraged to either replace the term background with the term aim or very briefly add a line on the background of the study.

We apologize for this error and according to the reviewer suggestion and the instruction for authors we add a line regarding the background. ‘Refractory glaucoma still represents a challenge for ophthalmologists to manage intraocular pressure’.

Methods section:

Methods section needs to be restructured:

Line 53: the heading experimental section is unfamiliar and needs to be replaced with Methods.

We agree with the reviewer, consequently the heading has been changed accordingly.

Lines 54-58: These lines should fall under the subsection Ethics Statement. Line 58: Institutional review board number is missing and needs to be added.

Lines 59-93: these lines need are not organized and need to fall under the subsections such as patient recruitment, surgical technique, data collection... Authors are encouraged to reformat this part of the methods.

We fully agree with this observation. The following subsection: study design and participants selection, ethics statement, data collection and outcomes measures, surgical technique and statistical analysis has been added as the institutional review board number.

Results section:

The comma in decimals values should be replaced with ‘.’ all over the text.

We really apologize for this error and we improve the manuscript as suggested changing comma with ‘.’.

Line 120: IOP -> needs to be replaced with its acronym.

We change IOP with intra ocular pressure as suggested.

Figure caption is missing; authors are encouraged to briefly describe the figure (box plots), the type of test used, etc.. Line 128: Idem as Figure 1. Line 185: Idem as Figure 1.

We thank the reviewer for the observation, and we modify the manuscript adding figure’s captions following the JCM recommendations.

Discussion Section:

Searching the literature identified several similar articles:

Safety and Efficacy of Ahmed Glaucoma Valve Implantation in Refractory Glaucomas in Northern Indian Eyes. Suneeta Dubey et al, 2015.

Surgical Outcomes of a Second Ahmed Glaucoma Valve Implantation in Asian Eyes with Refractory Glaucoma. Sze Chuan Ong et al, 2020

Authors need to compare their results to these articles and mention their novel and the added value of their current manuscript.

We thank the reviewer for these relevant indications; the first paper referred to AGV in refractory glaucoma patients so we add a sentence in the Introduction paragraph ‘Dubey and co-authors reported a success rate in Northern Indian eyes with refractory glaucoma of 85.45% at 1 year and 79.63% at 3 years after AGV implantation [8]’; we prefer not to compare our results with these one, because we presented success rate of a second AGV implant in refractory glaucomas.

Regarding the second manuscript we implemented our discussion comparing our results with those presented by the suggested Authors and we add a paragraph in the discussion.’Recently, Ong et al. published a report regarding the use of a second AGV implantation in Asian eyes with refractory glaucoma. Authors reported in this Asian population a failure rate of 9.5%, 20.0%, 32.5%, and 46% respectively at six months, one, two, and three years of follow-up; the reduction of IOP and medications was statistically significant at final follow up visit. These results are concordant with our findings in Caucasian descent. Still, again baseline IOP values of our population were higher (35.2± 7.9 mmHg vs 23.4± 7.9) and our criteria of success were more stringent compared to the one used by the authors. The first glaucoma drainage implant represents another difference because in our series all patients underwent AGV, instead in the paper published by Ong and co-authors patients also underwent Baerveldt implant in one case and in one patient it was not reported; moreover, looking at glaucoma types no cases of primary angle closure glaucoma was included despite the Asian race [30]. Finally, regarding visual acuity during follow-up, this group confirmed our findings, no significant difference in terms of BCVA was reported [30]’.

Line 164-167: this phrase is misplaced; authors are encouraged to move it to the results section.

We verified the reviewer suggestion; the sentence was already included in results section in the first version of the manuscript; table 2 as indicated by JCM guidelines should be presented near to the first time it is cited and divide this phrase from the rest of results paragraph.

Lines 235-237: the ID of the clinical trial needs to be added: NCT02691455.

We agree with the reviewer and we added the number as indicated.

Line 241: As mentioned in the limitation, the authors have conducted their study on a limited sample size... Did they try to carry out any sampling/experimental design before conducting the study? Determining the optimal sample size for their study could provide readers about the adequate number of participants needed to detect robust significant results!

The reviewer’s suggestion is very useful, post-hoc power analyses were performed to evaluate the sample sizes. The sample power value of the primary outcomes was > 0.8 (the minimum acceptable value) until 2.5 years. The sample size of the following period didn't reach the minimum power level. This is probably the reason why the p-value become less significant after 3 years. We consequently add a comment in the discussion paragraph.

Reviewer 2 Report

  1. In this retrospective, multi-center study, it would be helpful to the reader to know the denominator in terms of the total number of primary Ahmed valve implants were performed for refractory glaucoma during the 7 year time period (Jan 2011-Dec 2017). Thus, the reader can estimate the percentage of these primary Ahmed valve implant cases required second Ahmed valve implantation.

2. How many patients were excluded from the study population due to age less than 18 years and/or less than 3 months of follow-up after surgery?

3. Did the study surgeons consider placing a non-valved glaucoma implant (e.g. Baerveldt implant) at time of second implant surgery?

4. Did the study surgeons employ use of antimetabolites (e.g. subconjunctival injection of 5-FU postoperatively) to reduce fibrous encapsulation of the Ahmed implants to reduce need for second implant surgery?

Author Response

Comments and Suggestions for Authors

  1. In this retrospective, multi-center study, it would be helpful to the reader to know the denominator in terms of the total number of primary Ahmed valve implants were performed for refractory glaucoma during the 7 year time period (Jan 2011-Dec 2017). Thus, the reader can estimate the percentage of these primary Ahmed valve implant cases required second Ahmed valve implantation.

The observation of the reviewer is correct this could represent a useful information for the reader; all patients had at least one intervention before the first AGV implant while 22 patients (79%) had two glaucoma surgeries before the second AVG implant. No patient in our series did AGV as primary surgical intervention for glaucoma. We add a sentence in results and repeated this concept during discussion. ‘All patients had at least one surgical procedure before the first Ahmed Glaucoma Valve implantation and twenty-two patients (79%), before the second implant, had already two surgical glaucoma intervention. No patient did AGV as primary glaucoma surgery’.

  1. How many patients were excluded from the study population due to age less than 18 years and/or less than 3 months of follow-up after surgery?

We really appreciate this observation and we add a sentence in results regarding the mean age of our population (we didn’t exclude any patients for age < 18 years) ‘Twenty-nine patients (17 men and 12 women) who underwent two consecutive AGV implants during the study period were identified. One male patient was lost before three months of follow-up; the patient died for heart failure, so twenty-eight patients were finally enrolled. The mean age of our population was 59 years (range 26-79 years), any patient was excluded for age criteria.’

  1. Did the study surgeons consider placing a non-valved glaucoma implant (e.g. Baerveldt implant) at time of second implant surgery?

The reviewer observation is correct, we preferred to use valved tubes to avoid early postoperative complications.Both valved or non-valved glaucoma drainage implants have been used in patients with refractory glaucoma with a variable degree of success. Our population had a very high IOP at baseline, and there was a real risk of hyperfiltration and hypotony. For these reasons, in both centres devices with flow-restricting technology have been preferred.’

  1. Did the study surgeons employ use of antimetabolites (e.g. subconjunctival injection of 5-FU postoperatively) to reduce fibrous encapsulation of the Ahmed implants to reduce need for second implant surgery?

Based on patients’ chart there is no report regarding the use of antimetabolites during surgery or postoperatively. We prefer to add a sentence during surgical technique explanation to clarify this concept ‘Both surgeons avoid the applications of subconjunctival antimetabolites.’

 We also add a sentence in results when speaking about further surgical procedure ‘Further surgical interventions after AGV implantation in our series were: tube shortening (4 patients), capsule revision (3 patients) without the use of antimetabolites’.

Round 2

Reviewer 1 Report

The authors answered my comments.

Reviewer 2 Report

The edits are acceptable.